# OpenReview forum: "Rule-Guided Language Model Alignment for Text Generation Management in Industrial Use Cases"
_NeurIPS.cc/2024/Workshop/SafeGenAi — SafeGenAi Poster_

### Official Review · Reviewer_m3JL · 2024-10-08
**Evaluation of Rule-Guided Language Model Alignment for Enhanced Text Generation in Industrial Applications**

**Rating:** 4
**Confidence:** 3

**Review:**

This paper studies the challenge that  LLMs need to follow complex and numerous rules in industry applications. To solve the problems, the paper introduces a dataset of rule-applied responses to make LLM follow safety rules. The proposed scheme is technical and easy to follow, making it essential in industrial scenarios.
There are some problems, which must be solved before it is considered for publication.
First, in the proposed dataset, there appears to be a lack of corresponding metrics to evaluate its quality. This absence makes it challenging to assess the dataset's reliability and effectiveness.
Second, the llama2-13b-chat model has been available for some time, yet I haven't seen other models evaluated on this dataset for reference and comparison. This would be helpful for understanding the performance context.
Third, the contribution is not sufficiently clear; I am uncertain whether the primary focus is on the proposed dataset or the training method for the LLM. Clarifying this distinction would enhance the overall understanding of the work.
In conclusion, while the proposed approach offers valuable insights into rule-guided language model alignment, further clarity on the contributions and the inclusion of quality evaluation metrics would strengthen the work. Overall, this research has the potential to enhance the application of LLMs in industrial settings, and I look forward to seeing how it evolves.

---

### Official Review · Reviewer_ReyJ · 2024-10-08
**Review for Rule-Guided Language Model Alignment for Text Generation Management in Industrial Use Cases**

**Rating:** 6
**Confidence:** 5

**Review:**

This paper proposes a method to identify if a rule should be applied to a response. This paper shows results that prove that its approach improves the LLM performance compared to applying rules using simplified rule selection. Few comments below:
1. Can the authors elaborate why they chose Elo scores for comparison? Are there any other metrics?
2. Can the authors compare with the latest models, like mistral?
3. Please fix errors in paper, the author citations are incorrect.
4. Please fix grammatical errors, some words are uppercase in middle of sentence, like "we Revised the response"

---

### Official Review · Reviewer_bV7H · 2024-10-09
**Real-World Usage**

**Rating:** 7
**Confidence:** 4

**Review:**

This paper addresses the challenge of applying Large Language Models (LLMs) to industry-specific applications, wherein domain-specific rules must be adhered to. The proposed approach is centered on leveraging internal domain-specific rules to align the behavior of LLMs with the requirements of specialized tasks. The key method introduced consists of two steps:

1. creating a dataset of responses with partially applied rules using a simplified rule selection framework;
2. training the LLM on this dataset to generalize over the rules and correctly apply them in relevant contexts.

The approach is demonstrated in the automotive repair domain, where it significantly enhances the model's ability to follow safety rules when providing recommendations.

Strengths:

1. The paper presents a practical and resource-efficient approach to adapting LLMs to domain-specific scenarios without the need for large-scale post-training on specialized datasets.

2. Utilizing a simplified rule selection mechanism to generate rule-applied responses is a novel concept that addresses the complexity of managing numerous domain-specific rules.

3. The method enables the LLM to generalize across rules, even if the initial rule selection is not flawless, resulting in better performance in domain-specific tasks.

4. By concentrating on domains with natural language rules, the paper demonstrates the relevance and practicality of its approach in real-world industrial applications.